# SARS-CoV-2 Outbreaks on Mink Farms—A Review of Current Knowledge on Virus Infection, Spread, Spillover, and Containment

**DOI:** 10.3390/v16010081

**Published:** 2024-01-04

**Authors:** Mohammad Jawad Jahid, Andrew S. Bowman, Jacqueline M. Nolting

**Affiliations:** Department of Veterinary Preventive Medicine, College of Veterinary Medicine, The Ohio State University, Columbus, OH 43210, USA; jahid.1@osu.edu (M.J.J.); bowman.214@osu.edu (A.S.B.)

**Keywords:** severe acute respiratory syndrome 2, farmed mink, animal reservoir, public health

## Abstract

Many studies have been conducted to explore outbreaks of SARS-CoV-2 in farmed mink and their intra-/inter-species spread and spillover to provide data to the scientific community, protecting human and animal health. Studies report anthropozoonotic introduction, which was initially documented in April 2020 in the Netherlands, and subsequent inter-/intra-species spread of SARS-CoV-2 in farmed mink, likely due to SARS-CoV-2 host tropism capable of establishing efficient interactions with host ACE2 and the mink hosts’ ability to enhance swift viral transmission due to their density, housing status, and occupational contacts. Despite the rigorous prevention and control measures adopted, transmission of the virus within and between animal species was efficient, resulting in the development of mink-associated strains able to jump back and forth among the mink hosts and other animal/human contacts. Current knowledge recognizes the mink as a highly susceptible animal host harboring the virus with or without clinical manifestations, furthering infection transmission as a hidden animal reservoir. A One Health approach is, thus, recommended in SARS-CoV-2 surveillance and monitoring on mink farms and of their susceptible contact animals to identify and better understand these potential animal hosts.

## 1. Introduction to SARS-CoV-2

Coronaviruses (CoVs) are enveloped [1,2], single-stranded [2,3], positive-sense RNA viruses [1,2,3,4,5] belonging to the Coronaviridae family and Nidovirales order [3,4,5,6]. Four genera of CoVs have been recognized to date: Alphacoronavirus, Betacoronavirus, Gammacoronavirus, and Deltacoronavirus [3,5,7]. The first two predominantly infect mammals, while the last two genera primarily infect birds [2]. The virus name comes from its crown structure as visualized via electron microscopy [4]. CoVs are pathogenic in various mammalian and avian hosts, causing subclinical asymptomatic infections to clinical infections ranging from mild, to moderate, to life-threatening cases infecting the respiratory (upper and lower), gastrointestinal, and nervous systems [3,4]. CoVs can adapt to new hosts due to their propensity for high mutation and recombination rates [8]. During recent decades, CoVs have resulted in three major epidemics in human populations—SARS-CoV (severe acute respiratory syndrome coronavirus) in 2003, MERS-CoV (Middle East respiratory syndrome coronavirus) in 2012, and SARS-CoV-2 (severe acute respiratory syndrome coronavirus 2) in 2019 [9]—with the most recent resulting in a devastating pandemic.

On 21 December 2019, several human pneumonia cases with an unknown causative agent developed in Wuhan, China [5,6,10,11]. Patients had an epidemiological connection with the Huanan seafood market of Wuhan in Hubei province [5,6,11]. On 3 January 2020, next-generation sequencing of one patient’s respiratory sample revealed a novel human-infecting coronavirus [10], which was named 2019 novel coronavirus-infected pneumonia on 12 January 2020 by the WHO (World Health Organization) [3]. Later, the International Committee on Taxonomy of Viruses (ICTV) renamed the virus to SARS-CoV-2 because of its genetic similarity to SARS-CoV [11]. The disease was then officially named coronavirus disease 2019 (COVID-19), and the WHO declared a pandemic on 11 March 2020 [11]. As of 10 November 2023, there have been 697,623,269 human cases and 6,936,788 human deaths associated with SARS-CoV-2 reported globally [12].

SARS-CoV-2 is a 29.9-kilobase, positive-sense, single-stranded, RNA virus [6] causing respiratory diseases of varied severity [13]. It belongs to the Sarbecovirus subgenus of the Betacoronavirus genus [3,6,14] in the Coronaviridae family [3,5,6]. Betacoronaviruses include SARS-CoV, MERS-CoV, bat-SARS-like (SL)-ZC45, and bat-SL-ZXC21 [3]. SARS-CoV-2 has 96.1% genome sequence similarity to the bat-SL-CoV strain RaTG13 [15], 79.6% to SARS-CoV [14,15], and ~50% to MERS-CoV [6]. Nonetheless, among all the SARS-CoV-2-related CoVs, the highest sequence similarity is found between BANAL-20-52 and SARS-CoV-2 [16]. BANAL-20-52 was isolated from the Rhinolophus malayanus cave bat in northern Laos, Indochinese Peninsula [17], sharing nucleotide sequence identity of almost 96.8% with SARS-CoV-2 [16]. Additionally, the Spike (S) protein of BANAL-20-52 shows >98% amino acid sequence similarity with that of SARS-CoV-2, with 98.1% at the N-terminal domain (NTD) and 97.4% at the receptor binding domain (RBD) [16].

While there is a high similarity between SARS-CoV-2 and RaTG13, the RBD (receptor binding domain) of SARS-CoV-2 is highly divergent from that of RaTG13 [18]. Therefore, it is possible that due to selection pressure, the SARS-CoV-2 RBD, following its appearance in bats, further evolved in an unknown intermediate host and then zoonotically spread to humans [18]. Additionally, based on high similarity in the RBD of Malayan pangolin CoVs to that of SARS-CoV-2 [7,19,20], combined with evidence reporting vertical transmission of SL-B-CoVs in pangolins, it has been asserted that SARS-CoV-2 circulates in these animals [20]. Nonetheless, the absence of SARS-CoV-2 in 334 Sunda pangolins (*Manis javanica*) sampled over a decade from multiple locations in Malaysia before their illegal trade to other destinations suggests that the detection of SARS-CoV-2 in confiscated pangolins in China is likely due to exposure to other infected animals or humans [21]. Thus, Sunda pangolins are considered an incidental host but not a reservoir host for CoVs [21]. There is no conclusive evidence to support the hypothesis that pangolins or bats (RaTG13) are the natural reservoirs for SARS-CoV-2 [18].

It is asserted that the current SARS-CoV-2 pandemic has a zoonotic origin with a possible link to the fresh market in Wuhan, which sold numerous live animals including fish, shellfish, exotic animals, birds, and poultry [22]. SARS-CoV-2 is assumed to have crossed the species barrier and transmitted to humans through a reservoir host (possibly bats) via an unknown intermediate host [5,14,18,19,23,24,25,26,27,28]. However, the animal reservoir has not yet been identified [5,18,22,29,30].

## 2. Virus Transmission between Humans and Animals

CoVs utilize the spike (S) glycoprotein on their envelope to bind to cellular host receptors [31]. The RBD, otherwise known as the S1 C-terminal domain [15], of the S protein interacts with angiotensin-converting enzyme 2 (ACE2), which is found on the surface of various host cells [32]. This interaction mediates fusion of the viral membrane into the host cell, initiating infection [26,31]. Thereby, it is essential for SARS-CoV-2 to bind to host ACE2 to fuse into target cells and initiate infection [31]. A study of eight captive Malayan pangolins showed expression of ACE2 in multiple organs, including the kidneys (highest level of expression), heart, spleen, stomach, pancreas, lymph nodes, liver, and lungs, suggesting that all these organs can be infected by SARS-CoV-2 [28]. ACE2 from humans [15], cows, hamsters, and cats also interacts with high affinity with the SARS-CoV-2 RBD [26]. Conversely, ferrets, which are an important model for CoV infection, and horseshoe bats, which initially were thought to be a potential reservoir, have ACE2 proteins that poorly bind to the SARS-CoV-2 RBD [26].

Natural and experimental infections reveal the susceptibility of a variety of animal species to SARS-CoV-2 infection. Experimentally infected animals, including fruit bats [33], dogs [24], cattle [30,34], cats [24], goats [30], rabbits [30], ferrets [24,33], mink [35,36], hamsters [37], rhesus macaques [38], tree shrews [39], cynomolgus macaques [40], and African green monkeys [41], are susceptible to SARS-CoV-2 infection. Animal–animal transmission of SARS-CoV-2 infection through direct contact also has been shown in experimentally infected ferrets, cats, hamsters, and tree shrews [22,25]. Nonetheless, experimentally infected pigs [24,33], horses [30], chickens [24,33], and ducks [24] are not susceptible to SARS-CoV-2. Additionally, natural infection of SARS-CoV-2 has been detected in cats [42], dogs [43], tigers and lions [29], white tailed deer [44], feral/free-roaming mink [45,46,47], and recently farmed mink [13,22,48,49,50,51,52,53,54,55,56,57,58,59]. Further, zoonotic spread of SARS-CoV-2 from experimentally [25,48] and naturally [22,59,60] infected animals to susceptible humans has been reported, and anthropozoonotic transmission of SARS-CoV-2 from infected humans to susceptible animals (i.e., farmed mink [22,52,53,61] and certain zoo animals [62]) has also been reported.

## 3. Fur Farming Industry

Fur farming is the practice of selecting, breeding, and farming fur-producing mammals (e.g., fox, mink, coyote, rabbit, chinchilla, raccoon) for their skins (i.e., pelts) [63]. Mink and fox are the two dominant animals farmed for fur, with mink outnumbering fox [64]. The term mink typically refers to at least two distinct species in the Mustelidae family: the American mink (*Neovison vison*, the species we refer to throughout this review) and the European mink (*Mustela lutreola*) [65]. Since prehistoric times, people have hunted certain mustelids, including sable, ermine, and mink, for their fur [65]. While some mustelids are domesticated and kept as pets, the aggressive behavior of the mink has prevented them from being household pets, yet their lustrous fur has led to people raising them on mink ranches [65,66].

Mink have short, dense, and soft fur that is supplied to fashion and clothing companies for decorating collars, hoods, shoes, scarves, and other clothing [48,67]. Female mink are bred annually in March and whelp in May [66,68]. After weaning in July, kits are transferred to individual cages until they are pelted in fall and winter [65,66,68]. Farmed mink are fed using a commercially available, pumpable wet feed made from a mixture of animal byproducts, slaughter offal, and cereals [69].

In 2020, there were 120 mink farms in the U.S., with 90% of these located in Utah, Oregon, Idaho, Wisconsin, Minnesota, Michigan, Iowa, Pennsylvania, Washington, and Illinois [67]. These ranches produce 2 million pelts annually, with a total value of USD 80 million [70]. The European Union raises 34.7 million mink annually, with Denmark being the largest mink pelt producer in the world [69,70,71]. Following Denmark, China raises 20.7 million, the U.S. raises 3.1 million, and Canada raises 1.76 million mink annually [70].

By nature, these wild animals are not well-adapted to captive farm settings where they are kept in cages and cannot roam, hunt, hide, climb, or swim [64]. Fur animals raised under such conditions can experience behavioral diseases that may impact their immune systems, making them more prone to various illnesses [63]. The intensive management of mink farms, which keep animals with high genetic similarity, may accelerate the circulation and spread of diseases [63,65].

## 4. SARS-CoV-2 Outbreaks in Farmed Mink

Although human–human transmission is the main cause of SARS-CoV-2 spread, the number of SARS-CoV-2 infections in animal populations is also increasing [72]. As of 30 June 2023, there have been 775 outbreaks of SARS-CoV-2 in different animal species reported globally, affecting 29 species across 36 countries [73].

The recent SARS-CoV-2 outbreaks in animals (updated on 30 June 2023) were reported by Argentina in armadillo, Ecuador in spider monkey and woolly monkey, and Italy in American mink [74]. Outbreaks of SARS-CoV-2 in farmed mink were reported for the first time in the Netherlands on two mink farms in April 2020 [22,48]. Soon after, SARS-CoV-2 infections were reported in mink in Spain in May 2020 [49]; in Denmark in June 2020 [51,75]; in the U.S. in August 2020 [13]; in Sweden [76] and Italy [61] in October 2020; in Lithuania, France, and Greece in November 2020 [77]; in Canada in December 2020 [52,68]; in Poland in January 2021 [55]; and in Latvia in April 2021 [77] (Figure 1).

By 29 January 2021, SARS-CoV-2 infection had been documented in 400 mink farms across eight countries in the European Union: 290 farms in Denmark, 69 in the Netherlands, and the rest in France, Greece, Italy, Lithuania, Spain, and Sweden [78]. Human mink farm workers are reported as the probable source introducing SARS-CoV-2 infection to the farmed mink [22,48,75,78]. The American mink is the nonhuman animal species with the highest morbidity and mortality related with SARS-CoV-2 [13]. The American mink remains the only known animal species that has spread SARS-CoV-2 back to humans [63].

### 4.1. The Netherlands

As of 2020, there were 125 mink farms in the Netherlands producing 4 million mink and employing around 1600 full- and part-time staff [48]. On 19–20 April 2020, two mink farms located in the North Brabant province in southern Netherlands reported respiratory clinical signs in mink, coinciding with increased mortality [22,48]. On 23 April 2020, SARS-CoV-2 infection was confirmed in these two mink farms [22,48]. These were the first reported cases of SARS-CoV-2 infection in mink farms worldwide [48]. Confirmed cases of SARS-CoV-2-like illness were reported in farm workers prior to manifestation of the clinical signs in the farmed mink [48]. An analysis of viral sequences obtained from mink on these two farms indicated humans as the most likely source of the initial infection in the mink [48,59], which was followed by subsequent mink–mink [48,59], mink–human [48,59,71], and human–human [71] transmission of the mink-associated virus.

By 4 November 2020, 68 of the 126 mink farms (IDs: NB1–NB68) in the country reported SARS-CoV-2 infection [59]. The results of an initial analysis showed that mink sequences from the first 16 farms clustered into five discrete groups, indicating separate human introductions of the virus to the farmed mink [59]. It was demonstrated that farm workers contracted SARS-CoV-2 from mink rather than from humans in the same neighborhood, indicating that the farmed mink were the source of human infections on these farms [22,59]. Additionally, SARS-CoV-2 was detected on farm NB3 on 7 May, while SARS-CoV-2 testing was negative for all employees of the farm until 19–26 May, when five of the seven farm workers tested positive for SARS-CoV-2 [22]. Clustering of SARS-CoV-2 sequences derived from these five persons with those obtained from mink at farm NB3 showed that SARS-CoV-2 infection of the farm workers occurred after infection of the mink, indicating possible zoonotic spread of SARS-CoV-2 from farmed mink to humans [22,48,59]. Overall, zoonotic transmission of the virus from mink to humans was documented on at least 43 farms [59].

Whole-genome sequencing (WGS) of 34 SARS-CoV-2-positive human samples collected from within ~19 km^2^ of the four mink farms that initially reported SARS-CoV-2 infection (NB1–NB4) showed no relation between community-acquired and mink-associated SARS-CoV-2 clusters, thus indicating no spillover of SARS-CoV-2 from mink farms to the neighboring communities [22]. Nevertheless, instances of the virus jumping back and forth between humans and animals from different farms were documented, likely due to shared personnel, feed suppliers, and veterinarians between farms [59].

The infection control strategy for the Netherlands included strict hygiene protocols; banning of the transport of mink and their waste and products; and culling of all infected mink farms in the Netherlands within 6 days post-sampling, beginning in June 2020 [59]. No further SARS-CoV-2 infections were found on mink farms in the Netherlands after the last infected mink farm was documented in November 2020, likely because of the absence of farmed mink in these areas and the beginning of the harvesting season, when all mink, including adults, were harvested (pelted) because of the ban on mink farming effective January 2021 [59]. No additional human cases of SARS-CoV-2 associated with mink strains have been detected in the Netherlands since November 2020 [59].

### 4.2. Spain

Human cases of SARS-CoV-2 infection were detected in May 2020 in Teruel, Spain, when the wife of a mink farm worker tested positive for the virus [49,79]. PCR and serology confirmed the presence of SARS-CoV-2 on one mink farm, and humans were indicated as the probable source of introduction to the mink farm [49]. This was the first documented outbreak of SARS-CoV-2 in farmed mink in Spain, reported on 21 May 2020, involving almost 100,000 mink that were subsequently culled [49,79]. As a biosecurity measure, with the detection of SARS-CoV-2 on this farm, cleaning and disinfection of all the mink facilities were ordered [49]. During the outbreak, there was no indication of respiratory illness, anorexia, or high mortality in the infected farmed mink [1,49,79].

It is worth mentioning that during the first two years of the SARS-CoV-2 pandemic, no evidence of the virus infection was found in wild mink (*Neogale vison* and *Mustela lutreola*) in Northern Spain [80]. This finding is consistent with that of a British wildlife study [81].

### 4.3. Denmark

The first outbreak of SARS-CoV-2 in Denmark—the largest mink pelt producer in the world—occurred at three mink farms on 8 June 2020 [51]. From the beginning of the outbreak until the end of November 2020, 290 (25%) of the 1147 mink farms in Denmark reported SARS-CoV-2 infections [58,78]. The mink on one-third of the infected farms were asymptomatic, while those on the remaining farms showed clinical symptoms including anorexia, respiratory signs, and increased mortality [51]. Samples collected from the air (inside the farm close to mink snouts), mink fur, flies, feet of seagulls, gutter water, cats, and dogs that were present in/around some of the infected premises also tested positive for the virus [51], indicating transmission of the virus from mink to other susceptible animal species. However, samples taken from chickens, rabbits, and horses on some of the infected premises were negative for the virus [51].

Again, the SARS-CoV-2 outbreaks on these mink farms are believed to be due to spillover from humans, or reverse zoonosis [32,75,82]. The virus adapts well in the mink population, resulting in subsequent mink–mink and mink–human spread of mink-associated strains [58,75,78,83]. Of the 65,872 confirmed human cases of COVID-19, mink-associated strains were responsible for 895 (~6%) human cases across the North, Central, Southern, Zealand, and Capital regions of Denmark [58]. Of the mink-associated strains, Cluster 5 threatened public health, as it contains several mutations in the RBD of the S protein that can result in functional and/or structural alterations [9]. Cluster 5 was found in farmed mink on 5 farms and in 12 humans in the Northern Jutland area [9]. On 4 November 2020, the decision was made to cull all 17 million mink in Denmark and ban mink farming until end of December 2021 [32,54,58,78,83,84]. By 25 November 2020, all 290 virus-positive mink farms, along with all mink farms located within 7.8 km of infected farms, were culled, with the few remaining farms culled in December 2020 and January 2021 [58]. The incidence of mink-related strains infecting humans significantly decreased after the culling of the mink [54,78]. In February 2021, it was declared that Cluster 5 viruses are not circulating in humans in Denmark anymore [58].

### 4.4. Poland

Following the culling of mink in the Netherlands and Denmark, Poland became the largest mink producer in the European Union and second in the world, after China [55,83]. As of March 2023, Poland had 166 mink farms, whereas there were 350 farms before the COVID-19 outbreak in 2019 [56,83]. In response to the WHO, European Center for Disease Prevention and Control, and World Organisation for Animal Health recommendations for monitoring and surveillance of mink farms for SARS-CoV-2 infection, the Polish government ordered active monitoring and close surveillance of mink farms across Poland beginning in fall 2020 [55]. Accordingly, the first outbreak of SARS-CoV-2 on mink farms in Poland was detected on 30 January 2021 on one mink farm in North Poland as a result of testing 28 mink farms between November 2020 and January 2021 [55,56,83]. Another 13 SARS-CoV-2-positive mink farms were detected through July 2022 [56,85], and an additional 3 positive farms were detected between September 2022 and January 2023 [56]. Strict biosecurity measures were put in place, and farm workers, owners, and visitors were required to be tested for SARS-CoV-2 regardless of their health status [85].

The source is not clear for most outbreaks in Poland, except one outbreak that occurred in December 2021 that is possibly linked to human introduction [85]. However, possible zoonotic spread of the virus from infected mink to farm workers was also detected [83,85]. In September 2022 and January 2023, a cryptic variant of SARS-CoV-2 with ≥40 nucleotide changes was detected on two mink farms in close proximity [56]. It is assumed that the infection was introduced by an unknown animal reservoir, for neither farm workers nor farmed mink in the region tested positive for SARS-CoV-2 [56]. All SARS-CoV-2 outbreaks in mink farms in Poland were asymptomatic, and transmission of the virus between farms was not reported [85]. The Polish government mandated culling of all mink on a farm if there was a >10% mortality rate observed and/or if instances of zoonotic transmission from infected mink to humans were detected [56].

### 4.5. United States

In August 2020, SARS-CoV-2 outbreaks on mink farms were reported for the first time in the U.S. on two mink farms in Utah [1,13,86,87]. The farmed mink displayed severe respiratory symptoms, decreased appetite, and a high mortality rate (35%–55%) [13]. Positive cases of COVID-19 were also reported among people working on these farms [87,88]. As of 3 July 2023, there have been 18 SARS-CoV-2 outbreaks in farmed mink reported in the U.S. [47]. Mink farms reported confirmed cases of SARS-CoV-2 occurring through May 2021 in Wisconsin (n = 3), Utah (n = 10), Michigan (n = 1), Oregon (n = 1), and another anonymous state (n = 1) [47].

These outbreaks are assumed to be epidemiologically linked to infected humans working on the farms, as the infection is transmitted from infected humans to mink, followed by subsequent transmission between mink and from mink to other on-farm domestic animals [13,68,88]. Serological and molecular detection of SARS-CoV-2 in feral/free-roaming cats in or around infected mink farms was also positive, indicating possible spread of the virus from mink to these susceptible animals [89]. No additional SARS-CoV-2 outbreaks in farmed mink in the U.S. were reported through November 2020–May 2021 [47]. It is estimated that by November 2020, more than 15,000 farmed mink died due to SARS-CoV-2 in the U.S. [1]. In the states where SARS-CoV-2 infections were reported, the authorities did not declare culling (slaughter) of the animals at that time but decided to closely monitor the situation [1].

### 4.6. Canada

As of early 2021, 64 active mink farms were operating in seven provinces in Canada, breeding a total of 194,000 mink [90]. Following the detection of SARS-CoV-2 infection on mink farms in Europe [48,51,58], the government initiated active surveillance and monitoring for SARS-CoV-2 on mink farms in October 2020 [68]. Accordingly, following the detection of two COVID-19-positive human patients, combined with increased farmed mink mortality (<1.5%) [52,68], the first SARS-CoV-2 outbreak on mink farms in Canada was detected on 2 December 2020 [68]. The second SARS-CoV-2 outbreak in farmed mink was detected on 23 December 2020, through routine surveillance of deceased mink [68]. Increased mortality (~3%) and clinical signs consistent with SARS-CoV-2 were observed at this farm [68]. A third outbreak was detected on another mink farm on 14 May 2021, again through testing of deceased mink [52,68]. The strain that infected the mink was the same as that circulating in humans in the local community [52,68]. These were the only outbreaks of the virus reported on mink farms in Canada.

Following the detection of these outbreaks, strict biosecurity measures including the quarantining of infected farms, isolation of patients and their immediate contacts, improved disinfection protocols (for farm staff, vehicles, and products), use of personal protective equipment, animal transport bans, and employee vaccination were adopted to break the transmission chain [52,68]. Additionally, to track and detect any potential SARS-CoV-2 transmission from farmed mink to surrounding wildlife through feral cats or escaped mink, wildlife surveillance began in January 2021 in Canada in areas surrounding the three infected mink farms [91]. Nine species comprising 71 animals, 3 of which were escaped mink, were captured and tested for SARS-CoV-2 [91]. While none of the 68 wildlife animals tested were positive for SARS-CoV-2, the escaped mink were positive for the infection, indicating the potential for transmission and dissemination of the virus to other susceptible animals through the escaped mink [91].

## 5. Mink–Mink and Mink–Human Viral Transmission Dynamics

Mink is a highly susceptible animal species to SARS-CoV-2 [13,52,57,79,86,91,92], being classified as a high-risk animal contracting the virus from infected humans as well as from other infected mink [92]. Because farmed mink are housed in (wire) cages with bedding that can produce a lot of dust and are farmed in large groups, there is ample chance for virus spread on mink farms [57,63,78].

SARS-CoV-2 transmission can occur both through direct and indirect pathways [78]. Mink housed in cages separated by permeable partitions can spread the virus to susceptible neighboring animals through direct contact [48]. Direct contact with infected mink is also a risk factor for susceptible humans contracting SARS-CoV-2 infection [22]. Humans contracting SARS-CoV-2 from infected mink can spread the virus to other susceptible humans, causing community spread of the disease [71,88].

Indirect contact between infected and susceptible mink can also occur through infectious droplets produced by infected mink, fomites (e.g., bedding materials, feed), or air/dust contamination [48,53,78]. In a study by Buckland et al. [51], 22 samples taken within distances of 10 cm and 1–3 m from the snouts of mink across different mink farms were positive for SARS-CoV-2 when tested via RT-qPCR. Further, viral RNA was present and detectable in farm environments for >60 days [53]. Contaminated dust and droplets are the principle means of SARS-CoV-2 transmission between mink populations and also present an occupational risk for farm workers contracting the virus from infected mink [25,27,48]. Infected mink can also spread the virus to other susceptible animal contacts, including on-farm and/or free-roaming/feral dogs and cats [51,65,88], insects, and wild birds [51].

Although in most outbreaks between-farm transmission of the virus was not detected [48], even in areas where farms were in close proximity to each other [85], SARS-CoV-2 infections did spread through neighboring farms in Denmark via human contact with the mink [51].

## 6. Mink as a Reservoir

Mink can be a reservoir for SARS-CoV-2, transmitting the virus among themselves and serving as a source for spillover into humans, as was seen in the Netherlands [22]. Initially introduced by humans with COVID-19 to mink in the Netherlands [48], SARS-CoV-2 evolved and adapted in the mink host, allowing the virus to spread within this animal population while maintaining its ability to efficiently interact with human ACE2 for cell entry and infection [93]. This enabled the virus to adapt in mink and jump back into humans with mink-specific mutations [9,93].

Interaction between a virus and a species-specific virus receptor is an important factor in host tropism and establishes an important barrier to inter-species virus transmission [94]. The genome of the SARS-CoV-2 virus is prone to mutations, leading to the development of novel variants that occasionally cross species barriers [95]. Mutations in the S protein of the miSARS-CoV-2 strains can potentially influence the host tropism, virus transmission, pathogenicity, and sensitivity of the virus to vaccine-induced immunity [93], posing a public health concern.

On 5 November 2020, more than 800 human cases related to mink variants, encompassing a group of unique mutations that were not identified elsewhere, were reported in Denmark [58]. These variants were grouped into five clusters containing seven mutations [93]. Remarkably, the Cluster 5 variant, a unique miSARS-CoV-2 strain that was detected from 12 human patients, contains mutations in the S protein [58,84,93]. The Y453F mutation, situated in the RBD of this strain, enhances viral fusion by improving the interaction between the S protein of SARS-CoV-2 and the ACE2 receptor of mink [93]. This enhances the co-circulation capability of miSARS-CoV-2 between mink and human hosts. Y453F demonstrates lesser sensitivity to neutralizing antibodies from convalescent serum from COVID-19 patients, thus suggesting its potential to escape the protection induced by vaccines [84,93].

Hence, viruses spread rapidly among mink populations, undergo mutations, and lead to the emergence and spread of novel mink-associated variants. Mink-associated variants can circulate among mink populations and spill over to other susceptible animals as well as susceptible humans, causing community spread of the virus through subsequent human–human transmission [22,71,75]. The chance of the general population contracting miSARS-CoV-2 variants is estimated to be low, yet the risk is moderate for populations living in the neighborhoods of mink farms and extremely high for those working on mink farms [76,96]. A study by Dall Schmidt and Mitze (2022) found that municipalities in Denmark that reported outbreaks of SARS-CoV-2 on mink farms showed up to 75% higher incidence of SARS-CoV-2 infection in humans compared to municipalities where outbreaks were not detected on mink farms [54].

## 7. Control Measures and Biosecurity

Soon after the detection of human cases of COVID-19 and their follow-up anthropozoonotic cases in farmed mink, the World Organisation for Animal Health (OIE) and other public health authorities around the world published guidelines aiming to prevent and control the intra-/inter-species transmission dynamic of SARS-CoV-2. The OIE published guidelines on minimizing the risk of SARS-CoV-2 spillover from infected humans to domestic animals, underscoring the implementation of risk reduction strategies, as well as monitoring and surveillance programs focusing on the susceptible animals based on a One Health approach [92]. USDA-APHIS published response and containment guidelines, focusing on the awareness of farm workers (education), implementation of biosecurity measures (minimum biosecurity plan), early detection of the virus (surveillance and monitoring), and eradication of the susceptible/infected farmed mink (elimination of the virus) [97]. Furthermore, the European Food Safety Authority (EFSA) and European Centre for Disease Prevention and Control (EUCDC) published a scientific report on monitoring SARS-CoV-2 infection in mustelids, considering all the mink farms at risk of SARS-CoV-2 infection, underscoring early detection of the virus through both active and passive monitoring and surveillance programs [78]. Additionally, the Canadian COVID-19 One Health Working Group published a guideline on managing SARS-CoV-2 in farmed mink, focusing on the prevention and detection of and response to SARS-CoV-2 infections on mink farms [90].

Having said that, mink farms were tested for SARS-CoV-2 when (i) a person who had a connection with the farm tested positive for COVID-19 [48]; (ii) clinical signs compatible with COVID-19 were seen in mink [48,51]; (iii) significant mortalities were observed on the mink farm [48]; (iv) SARS-CoV-2 was isolated from dead mink through surveillance programs [51]; (v) there was any link between a susceptible farm and an infected farm [51]; and (vi) as part of early monitoring programs for SARS-CoV-2 based on outbreaks detected in other countries [85].

Immediately after the detection of a SARS-CoV-2 outbreak, control measures were adopted across mink farms to contain further transmission, spread, and outbreaks of the virus. On-job awareness training for farm workers and farm owners was delivered by public health authorities to increase knowledge about SARS-CoV-2 and containment measures at both the personal and farm levels [88,97]. Enhanced personal protective equipment (i.e., gloves, face masks, face shields, coveralls, and hoods) were mandated for farm workers and owners when entering farms [68,88,97]. Biosecurity and control measures included the vaccination of farm workers [68,97], rigorous isolation and testing of animals, bans on transport of the animals as well as their waste and products, bans on farm visitors, implementation of surveillance programs combined with strict hygiene measures, and culling of mink on infected farms [59,85]. However, biosecurity measures implemented on mink farms in the affected areas (particularly in the Netherlands and Denmark) were found to be insufficient, as the virus continued to spread widely and rapidly among mink, and spillback events to people working on the farms or connected with the farms also continued [51,68].

Following animal testing as a control measure in some areas (e.g., Poland), mink farms were culled when the mortality rate of mink on the farm was >10% and/or when zoonotic transmission of the virus was detected from mink to humans [56]. In other areas (e.g., Netherlands), infected farms were culled if SARS-CoV-2 infection was detected there, regardless of the morbidity rate [48]. In Denmark, when culling only infected premises did not help to stop the SARS-CoV-2 outbreak, all farms, regardless of their infection status, were culled [51]. Culling of infected mink farms significantly reduced the incidence of human COVID-19 in the infected area [54].

## 8. Conclusions

Mink are farmed in small cages, in high densities, and under conditions that can affect their immune system, facilitating the contraction and spread of disease. Upon introduction, a SARS-CoV-2 infection can spread swiftly on a mink farm, eventually affecting most, if not all, of the mink. Mink are the only known animal species found to spread SARS-CoV-2 to humans and other contact animals.

The first confirmed anthropozoonotic, human–mink, spread of SARS-CoV-2 was documented in the Netherlands, occurring in farmed mink. Following the virus’s introduction to farmed mink, there was quick intra-/inter-species transmission of SARS-CoV-2, which resulted in millions of mink being culled around the world. While infections in farmed mink were initially detected by observing severe respiratory symptoms and increased mortality, many infections were subclinical with no apparent clinical signs, making mink hidden animal reservoirs/sources for SARS-CoV-2. Massive and progressive mink–mink transmission of the virus led to the emergence of persistent infections in the mink population, resulting in the emergence of strains with mink-specific mutations resistant to neutralizing antibodies and thus challenging public health. Spillback events of these mink-specific strains to humans and subsequent human–human transmission leading to community spread of the virus have been reported.

In the effort to preclude further inter-/intra-species transmission of the virus and control outbreaks, strict biosecurity and biocontainment measures were adopted. Nonetheless, the measures were insufficient, as the virus continued to spread not only among mink but also from mink to other susceptible animals and humans. As observed in Denmark and many other affected places, the culling of farmed mink could have a role in reducing the incidence of epizootic and zoonotic events of the virus. Nonetheless, this causes a catastrophic economic loss for the mink industry [71].

Various studies have been conducted in the affected fur-producing countries to investigate the intra-/inter-species transmission dynamics of SARS-CoV-2 in farmed mink. Thinking of the triad of diseases in a broader array and in the case of SARS-CoV-2 with its zoonotic, anthropozoonotic, and epizootic capability, comprehensive studies were undertaken in most of the affected areas, including a broader range of domestic and wild animal contact species, human contacts (both on and around the farms in the community), and varied types of human, animal, and environmental samples. These studies have made a huge contribution to the current knowledge we have on SARS-CoV-2 in this potent animal species—farmed mink.

SARS-CoV-2 possesses a high mutation rate, broad host tropism, and high spillover potential. This necessitates the application of a comprehensive strategy to study its epidemiology and transmission dynamics to not only control its further spread and spillover within and between animal–human populations but also provide sufficient knowledge to the scientific and policymaking communities to better prepare, prevent, and respond to future epidemics/pandemics. Current knowledge thus recommends a One Health approach in the surveillance and monitoring of SARS-CoV-2 in farmed mink. A One Health approach will ensure multisectoral collaboration between planners and policymakers, scientific bodies, public health authorities, the fur industry, and public communities responding to this One Health threat.

## Figures and Tables

**Figure 1 viruses-16-00081-f001:**
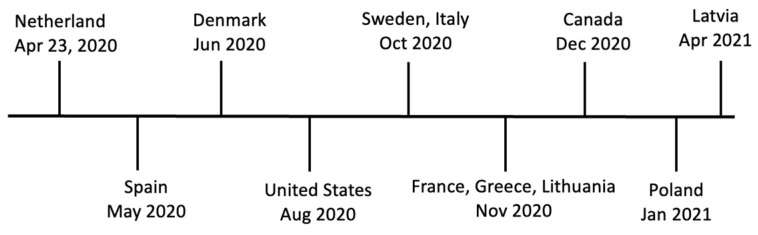
Timeline of SARS-CoV-2 infections on mink farms.

## Data Availability

Not applicable.

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
