# Peer review of "SARS-CoV-2 Outbreaks on Mink Farms—A Review of Current Knowledge on Virus Infection, Spread, Spillover, and Containment"

_viruses, 2024, doi:10.3390/v16010081_

Round 1

Reviewer 1 Report

Comments and Suggestions for Authors

This manuscript describes the summary of SARS-CoV-2 outbreaks on mink farms particularly on virus infection, spread, spillover and containment. Many studies have been conducted in several countries to explore outbreaks of SARS-CoV-2 in mink farms and its intra/inter-species spread including anthropozoonotic report. The paper comprehensively presents the most important aspects of SARS-CoV-2 infection on mink farms taking into account public and animal health.

However, in part 7, Control measures and biosecurity,  there are missing formal documents, legislation reports like European Union (EU) Commission Decision 2021/788 or WOAH, EFSA scientific reports on SARS-CoV-2 infections in mink that govered SARS-CoV-2 monitoring in Mustelidae.

I have noticed also few minor points that should be improoved:

Line 107: double "farming" , should be deleted

Line 147 and line 240: in my opinion reference 55 should be replaced by 54

Line 159: Actually data concern 2020 year, I think it should be remarked

Line  248 and after in section 4.4:  relevant information is missed:

The next three positive mink farms were identified between September 2022 and January 2023. The vast majority of mink SARS-CoV-2 infections were traced back to variants dominant in human samples during the COVID-19 epidemic in Poland, although one of the Alpha variants was detected many months reaching almost 2 years after the epidemic’s peak.

Line 259: The stentence should be corrected:

Polish legislation mandated the culling of all animals on a farm if the mink mortality rate exceeds 10% or in case of mink-to-human transmission. Mink were culled only in the only one farm [54].

Some references are not properly edited: missing year of publication bolded (ref?: 28, 29), missing year of publication (ref: 17, 21). I reccommend to check references before publishing.

Reviewer 2 Report

Comments and Suggestions for Authors

There have been other secondary zoonotic transmissions from animals to humans, beyond mink, as incorrectly stated. Some definite, other epidemiologically probable. The first include the Hong Kong outbreak from imported infected rats from Netherlands. The second include certain zoo secondary transmissions from animals to workers- a review can be found for example in Microorganisms journal, article "SARS-CoV-2 as a zooanthroponotic infection, spillbacks, secondary spillovers and their importance". There are numerous other reviews that include all these events, and of course the primary literature to all of them. 

2. A comment on the lack of detection of the virus in free roaming mink, in Spain and UK, might be useful, to add to the whole picture. 

3.  Since you discuss the close relatives of SARS-CoV-2, including ZXC45 and 21, to be accurate you should predominantly mention the BANAL betacoronaviruses from Laos, described in Nature in February 2022 by Temmam et al, they are the closest relatives. 

4. when discussing the Y453F mutation, one should be more specific. It didn't evade vaccine induced immunity, because there were no human vaccines back then, it was studied in a mice vaccinated model. And its overall effect was on evasion of cellular and not humoral immunity.  

Reviewer 3 Report

Comments and Suggestions for Authors

Mohammad Jawad Jahid and co-authors have prepared a very interesting review. The review is devoted to the analysis of the SARS-CoV-2 outbreak on mink farms. Indeed, since the beginning of the COVID-19 pandemic, there have been reports of similar outbreaks, first from the European region and then from other parts of the world.

The authors have collected similar information that reflects the events that occurred. The work is big and necessary. But it is not without its shortcomings. So I have some small comments:

1. It would be good to make a map that shows the regions in which outbreaks were recorded on mink farms.

2. It would be nice to make a table and show a comparison of the nucleotide sequences (in general) of isolates from minks and in the same region from humans.

3. It would be a good idea to try to find a connection between the size of the outbreak in humans in the region and in farmed minks.
